# Functional Enhancement of Guar Gum−Based Hydrogel by Polydopamine and Nanocellulose

**DOI:** 10.3390/foods12061304

**Published:** 2023-03-18

**Authors:** SolJu Pak, Fang Chen

**Affiliations:** College of Food Science and Nutritional Engineering, National Engineering Research Center for Fruit and Vegetable Processing, Key Laboratory of Fruit and Vegetables Processing Ministry of Agriculture, Engineering Research Centre for Fruits and Vegetables Processing, Ministry of Education, China Agricultural University, Beijing 100083, China; lb20193060027@cau.edu.cn

**Keywords:** hydrogel, gelatin−polydopamine, nanocellulose, guar gum, borate, self−healing

## Abstract

The development of green, biomedical hydrogels using natural polymers is of great significance. From this viewpoint, guar gum (GG) has been widely used for hydrogel preparation; however, its mechanical strength and adhesion often cannot satisfy the biomedical application. Therefore, in the present study, gelatin and a cellulose nanocrystal (CNC) were first applied to overcome the defects of guar gum hydrogel. Dopamine was self−polymerized into polydopamine (PDA) on the gelatin chain at alkaline condition, and gelatin−polydopamine (Gel−PDA) further cross−linked with guar gum and CNC via the borate−didiol bond, intramolecular Schiff base reaction, and Michael addition. CNC not only interacted with guar gum using borate chemistry but also acted as a mechanical reinforcer. The obtained Gel−PDA+GG+CNC hydrogel had an excellent self−healing capacity, injectability, and adhesion due to the catechol groups of PDA. Moreover, dopamine introduction caused a significant increase in the anti−oxidant activity. This hydrogel was cyto− and hemo−compatible, which implies a potential usage in the medical field.

## 1. Introduction

Hydrogel is a semi−solid and soft polymer network that can retain much water and keep its shape well. According to gelation chemistry, hydrogel can be classified into covalent hydrogels and noncovalent hydrogels. Covalent hydrogels can be formed by covalent bondings, such as EDC coupling, click reactions, Michael additions, disulfide bonds, and free−radical polymerization. Electrostatic interactions (mainly by multivalent cations, such as Ca^2+^ and Fe^3+^), polymer–polymer interactions, hydrophobic interactions, and π–π interactions (dextran or gelatin) are the main force to form noncovalent hydrogels. Meanwhile, hydrogel formation can be initiated directly or indirectly; so, intrinsic and indirect triggers are defined. Intrinsic triggers, the change of pH or temperature, components mixing, and enzyme addition can directly modify the polymer property or accelerate the cross−linking to initiate the gelation. Indirect triggers include temperature, ultrasound, and electromagnetic radiation which stimulates the cross−linker (cargo) release from carriers or activates the photo−crosslinker [1].

Hydrogel is known to form a physical or chemical barrier to microbial invasion and supply a relatively wet environment around the applied sites [2]. Because of its high hydrophilicity and biodegradability mechanical strength, hydrogel has been remarked as a novel material in biomedical fields including drug delivery [3], bone regeneration [4], wound dressing [5,6], and biosensors [7]. However, the lack of injectability and self−healing ability limits the broad medical application of traditional hydrogels [8]. On the other hand, many biomedical hydrogels have adopted various synthetic polymers, including polyvynyl alcohol (PVA), polyacrylic acid (PAA), and polyacrylamide (PAM), and undesired side effects might be present after application [9,10]. Therefore, hydrogel fabrication from natural and biocompatible polymers has gained great interest all over the world.

Guar gum (GG) is a natural polysaccharide isolated from the seed of a legume. Its main chain consists of linear 1, 4 linkages of β−D−mannopyranose, while the branch is composed of 1,6 linkages of α−d−galactopyranose. With abundant hydroxyl groups, guar gum has been applied for the facile fabrication of adhesive and stimuli−responsive hydrogels. Among cross−linking chemistry for guar gum−based hydrogel, the borate−didiol interaction was investigated, leading to the promising functional properties [11]. Nonetheless, the instability and weakness of neat guar gum hydrogel restrained its wide application. Actually, borate−based guar gum hydrogel showed a change in mechanical strength between 25 °C and 37 °C, which is a dissatisfying property for wound healing or cell scaffolding [12]. It was known that galactomannan is sensitive to temperature−based degradation [13]. The rheological value of guar guam solutions (storage modulus and loss modulus) decreased with the storage time, which probably means the depolymerization of macromolecules [14]. Moreover, taken with the reversibility of the borate ester bond, guar gum hydrogel still remains unstable with low strength. In this regard, some researchers tried to overcome this defect through the involvement of other polymers, such as chitosan and PAA [15]. Gelatin, a promising natural protein with various reactive functional groups, is renowned for its good gel−forming capacity. However, gelatin hydrogel obtained by unfolding or other physical cross−linking is quickly dissolved and mechanically weak so it cannot meet the practical requirement. Miscellaneous chemistry including a cross−linker (genipin or metal [16]) and grafting (thiol or amino grafting [17,18]) enables the formation of stable, tough, and stimuli−responsive hydrogels from gelatin. Recently, dopamine inspired by mussels has opened many possibilities to design and synthesize the novel hydrogels with excellent functional properties. Catechol groups of dopamine can endow the hydrogel with good tissue adhesion and self−healing ability [19]. Up to date, two major dopamine chemistries have been introduced for the modification of hydrogel precursors. The first chemistry−grafting is normally performed via the covalent bond between polymers and dopamine, and the most representative method is a (dimethylamino) propyl)−3−ethylcarbodiimide hydrochloride (EDC)/N−hydroxysuccinimide (NHS) coupling reaction [20,21]. Dopamine chemistry coating is more facile way to modify the polymer or metal surface [22]. To the best of our knowledge, surfaces including cells and tissues can be decorated with dopamine directly. A catechol of dopamine can be easily transferred into quinones in an alkaline environment (pH > 8.5). Then, it can be sequentially self−polymerized to form a polydopamine on the substrate surface. Similarly, tannic acid coating on the nanocellulose was reported to participate in the tough and adhesive hydrogels [23,24]. However, dopamine coating chemistry has not been extensively investigated for different polymers, and its functional and biological effect still remains unclear. Nanocellulose, such as a cellulose nanocrystal (CNC) and a cellulose nanofiber (CNF), is one of the versatile and sustainable polysaccharides derived from the most abundant resources—cellulose. The good biocompatibility and mechanical property has enabled its wide application for functional hydrogels through a Schiff base reaction, borate−didiol bond, and ionic interaction with other polymers such as PVA, alginate, and gelatin [25,26].

From the above viewpoint, we aimed to prepare an environment−friendly hydrogel mainly based on the guar gum, which will be utilized as wound dressings or tissue scaffolds. The adhesion and anti−oxidation of guar gum hydrogel were significantly improved with the introduction of the gel−polydopamine complex, which is easily produced. CNC can also contribute to the mechanical reinforcement of guar gum−based hydrogel. The obtained Gel−PDA+GG+CNC hydrogel showed a good cytocompatibility and hemolysis ratio, suggesting great potential for biomedical application.

## 2. Materials and Methods

### 2.1. Materials

Gelatin (Type A, 200 g bloom) and dopamine were purchased from Sigma−Aldrich (Shanghai, China). A cellulose nanocrystal (CNC: obtained by sulfuric acid hydrolysis, the degree of polymerization is 240) was supplied by Tianjin Woodelf Biotechnology Company (Tianjin, China). Guar gum (viscosity: 2500–3000 mPa·s) was granted by Beijing Biotopped Science & Technology Company (Beijing, China). Borate (Na_2_B_4_O_7_·10H_2_O), potassium bromide (KBr), and sodium hydroxide were provided by Sinopharm Chemical reagent Company (Shanghai, China). Phosphate buffer saline (PBS, pH 7.2–7.4), saline solution (0.9% NaCl), 1, 1−diphenyl−2−picrylhydrazyl (DPPH) free radical, and cell counting kit−8 (CCK−8) were procured from Solarbio Science & Technology Company (Beijing, China).

### 2.2. Synthesis of Gel−PDA

The Gel−PDA complex was synthesized from gelatin and dopamine at alkaline conditions [27]. Briefly, 3.5 g of gelatin was fully dissolved in 35 mL of distilled water at 40 °C under stirring, and pH was adjusted to 8.5 with a 5% sodium hydroxide solution. A total of 500 mg of dopamine hydrochloride was added into the gelatin solution and stirred slowly at 40 °C for 6 h to perform the oxidative polymerization of dopamine on the gelatin chain. The obtained gelatin−polydopamine complex (Gel−PDA) was freeze−dried at −60 °C and further used for its characterization and hydrogel fabrication. The Gel−PDA synthesis was performed in triplicate.

### 2.3. Characterization of Gel−PDA

#### 2.3.1. Ultraviolet−Visible (UV–Vis) Spectroscopic Analysis

An ultraviolet–visible (UV–Vis) spectrometer (Beijing Persee General Instruments Co., Ltd, Beijing, China) was used to confirm the PDA formation on the gelatin [28]. Briefly, the as−prepared Gel−PDA solution was diluted to 0.1% consistency with distilled water, while 0.1% gelatin solution was also prepared. Full scanning within the range of 200–700 nm wavelength was performed for gelatin and the Gel−PDA solution. The analysis was repeated three times for each sample.

#### 2.3.2. Fourier−Transform Infrared Spectroscopic Analysis

Fourier−transform infrared spectroscopy (Thermo Nicolet Corporation, Madison, WI, USA) was applied in order to characterize the PDA coating’s effect on the gelatin matrix [27]. Freeze−dried Gel−PDA was pulverized finely with a fully dried KBr crystal (mass ratio of Gel−PDA:KBr = 1:100) under an infrared ray lamp. Consequentially, this powder mixture was shaped into a thin disc to perform the analysis, which ranged from 400 to 4000 cm^−1^. Pure gelatin was also analyzed as a control. All procedures were repeated three times for each sample.

#### 2.3.3. X-ray Diffraction (XRD) Analysis

X-ray diffraction (XRD) analysis (Bruker Corporation, Billerica, United States) was further performed in the range of 2 θ = 10–80° at 5° min^−1^ speed to elucidate the possible change of gelatin crystallinity by dopamine polymerization [29]. Lyophilized Gel−PDA and pure gelatin were used for the XRD analysis. The experiments were performed in triplicate for each sample.

### 2.4. Fabrication of Gel−PDA+Guar Gum+CNC Hydrogel

The Gel−PDA+GG+CNC hydrogel was prepared following the protocols of [12] with a slight modification. A total of 60 mg of freeze−dried Gel−PDA was dissolved in distilled 5 mL of water at 40 °C, and 30 mg of CNC powder (obtained by the freeze−drying of neat CNC 2.5% dispersion) was dispersed in the Gel−PDA solution homogeneously under stirring (500 rpm) for 30 min followed by ultrasonication for 30 min. Afterward, 80 mg of guar gum powder was fully dissolved in the Gel−PDA+CNC mixture dispersion at 45 °C, in which 350 uL of 2% borate solution was dropwise added under stirring to form the hydrogel. Then, the hydrogel was further stabilized at 37 °C for 15 min. Additionally, Gel−PDA+GG, GG+CNC, and neat GG hydrogel were fabricated as a control.

### 2.5. Characterization of Hydrogel

#### 2.5.1. Morphological and Structural analysis

The morphological profile of hydrogels (Gel−PDA+GG+CNC and GG) was analyzed with a scanning electron microscope (Hitachi High−Technologies Corporation, Tokyo, Japan) at 10 kV, and 100 μm resolution photos were taken. Freeze−dried hydrogel pieces were cut by scissors to reveal the proper cross−section. The piece was fixed on a copper board and sprayed with gold [30]. FT−IR analysis was adopted to illustrate the functional groups of the lyophilized Gel−PDA+GG+CNC, Gel−PDA+GG, GG+CNC, and neat GG in the range of 400–4000 cm^−1^ at a resolution of 4 cm^−1^ [18].

#### 2.5.2. Rheology and Self−Healing Test

The rheological property of hydrogel was assessed using a TA rheometer (Waters Co., Milford, MA, USA) [31]. All the measurements were performed using a 40 mm parallel plate with a 1000 μm gap at 25 °C. Frequency sweep was performed within the angular frequency range of 0.1–100.0 rad s^−1^ at a 1.0% strain level, whereas 10.0 rad s^−1^ and 1.0–1000.0% were set as angular frequency and strain range, respectively. Storage modulus (G′) and loss modulus (G″) were recorded to evaluate the hydrogel rheology. For the macroscopic self−healing test, hydrogel samples were cut into two pieces and put together without any additional stimulus. After 10 min, pictures were captured to visualize the self−healing property of hydrogel.

#### 2.5.3. Tissue Adhesive Test

The tissue adhesive property of hydrogel was measured with wet pig skin using an Instron−5943 mechanical tester (Instron corporation, Canton, United states) [31]. The fresh pig skin without extra fat was purchased from a local market and cut into rectangular pieces (10.0 mm × 50.0 mm). Before the test, pig skin pieces were cleaned using saline solution (0.9% NaCl) and immersed in PBS (pH 7.4) at 10 °C for 12 h to keep the surface wet. A total of 200 μL of as−prepared hydrogel was injected into a 15 mm × 15 mm area of the pig skin piece, on which another piece was put to adhere to each other.. Sequentially, these pieces were immediately pressed together using a constant force and pulled apart with a 50 N load cell at a gauge length of 50.0 mm, as well as a cross−head speed of 5.0 mm min^−1^. The adhesive strength (Pa) was calculated by dividing the maximum load (N) by the cross−linked area (m^2^). The analysis was performed three times for each hydrogel to obtain the average value.

#### 2.5.4. Anti−Oxidant Activity Assay

A 1,1−diphenyl−2−picrylhydrazyl (DPPH) radical was used to evaluate the hydrogel anti−oxidant capacity [32]. A total of 100 mL hydrogel was added into a centrifuge tube containing 1 mL 200 μM DPPH solution (in ethanol) and was shake−incubated mildly at 30 °C for 1 h in the dark. Then, 100 μL of supernatant was transferred into a 96−well microplate to measure the absorbance at 517 nm using a microplate reader (Thermo Fisher Scientific, Waltham, MA, USA). The DPPH solution was used as a control, and scavenging capacity was calculated based on the following equation:DPPH scavenging % = (A_0_ − A_1_)/A_0_ × 100%
where A_0_ is the absorption of the DPPH solution in ethanol, while A_1_ is the absorption of the hydrogel sample+DPPH ethanol solution.

#### 2.5.5. Hemolysis and Hemostasis Assay

Sprague Dawley (SD) rats (female, 7–8 weeks, 220–240 g) were obtained from Charles River Laboratories Inc. (Wilmington, United States) and raised in a fixed environment (12 h, light/dark cycle, 25 ± 2 °C, 55% ± 10% humidity) so that they can obtain free access to water and feed. Animal experiments were governed by the Animal Protection Professional Committee of China Agricultural University. After adaptive feeding for 1 week, SD rats were anesthetized with chloral hydrate, and the fresh blood was taken into the centrifuge tube. The erythrocytes were immediately separated from the blood by centrifugation at 1500 rpm for 10 min and rinsed with PBS three times. Finally, 5% erythrocyte suspension was prepared by dilution with PBS. A total of 100 μL of hydrogel was added into a 2 mL centrifuge tube containing a 500 μL erythrocyte suspension, and was then shake−cultured at 37 °C at 100 rpm. PBS and 0.1% triton were used as the blank and positive control, respectively. After 1 h, the culture was centrifuged at 1500 rpm for 10 min, and 100 μL of supernatant was taken into a 96−well microplate to measure the absorbance at 540 nm using a microplate reader (Thermo Fisher Scientific). The hemolysis ratio was calculated using the following equation:Hemolysis ratio (%) = (A_s_ − A_b_)/(A_p_ − A_b_) × 100%
where A_s_, A_b_, and A_p_ are the absorbance values for the erythrocyte suspension with samples, PBS, and 0.1% triton solution, respectively [33]. All experiments were repeated three times.

For the hemostasis assay, the abdomen of an anesthetized rat was cut to reveal the liver, and filtration paper and plastic film were placed carefully below the isolated liver to prevent body fluids penetration. Bleeding was caused by cutting the liver with scissors, and then hydrogel (300 μL) was quickly applied for the hemostasis. The blood weight on the filtration paper was measured when no more bleeding was shown [34]. All experiments were repeated three times.

#### 2.5.6. Cytotoxicity Assay

Hydrogel cytotoxicity was measured following a similar protocol to [33,35]. A human hepatocellular carcinomas (HepG 2) cell was procured from the Cell Bank of the Chinese Academy of Sciences (Shanghai, China) [35]. The hydrogel samples were exposed to UV radiation for 30 min for sterilization, and then immersed in a DMEM medium (containing 10% fetal bovine serum, 100 unit/mL penicillin, and 100 μg/mL streptomycin) for 24 h at 37 °C. For the cytotoxicity experiment, a vial of frozen HepG 2 cells was thawed and seeded on 25 cm^2^ flasks. The cells were incubated in a DMEM medium (containing 10% FBS and 1% penicillin/streptomycin) up to 80% fullness and treated with trypsin. The HepG 2 cells were seeded in a 96−well plate at a concentration of 1 × 10^4^ (100 μL), and then the plate was further incubated for 18 h at 37 °C to promote stable cell attachment. Consequently, the culture medium was replaced with 100 μL of hydrogel extract to incubate at 37 °C, whereas untreated cells were a positive control. After 48 h, the culture medium was removed and rinsed twice with PBS. A total of 100 uL of a fresh DMEM medium, including a CCK−8 reagent (10 μL), was added to cell−attached wells and blank wells (blank), and then the plate was kept in the dark for 1 h at 37 °C. Finally, the absorbance of the solution was analyzed at 450 nm using a microplate reader (Thermo Fisher Scientific, Waltham, USA). Each sample had a six−well treatment in parallel.

### 2.6. Statistical Analysis

The data of all experiments were expressed as the average ± standard deviation. Statistical analysis was performed by one−way ANOVA. Symbols *, **, and *** correspond to a significance level of 0.05, 0.01, and 0.001, respectively.

## 3. Results and Discussion

### 3.1. PDA Coating on the Gelatin Matrix

Dopamine with an active catechol group is easily oxidized at the alkali condition and further polymerized to form a polydopamine, and this in situ polymerization is one of the promising and facile strategies to modify the polymers [19,36,37]. The gelatin solution was transparent at the beginning; however, it turned light pink and finally became black as the polymerization time increased. The UV−Vis spectrum showed an intrinsic peak at 230 nm for both gelatin and Gel−PDA, while a weak peak at 280–290 nm of gelatin was intensified at the Gel−PDA spectra, which means that there was PDA formation on the gelatin chain [27,38] (Figure 1a). Actually, both gelatin and dopamine have aromatic moieties; therefore, their complex might result in the overlapping of the π–π transition peak. Additionally, a weak peak shown around 420 nm is identified as the representative peak of PDA synthesis on the gelatin, which is time−dependent [28,39]. It was reported that dopamine polymerization might cause the broad alteration ranging from 300–600 nm at the gelatin spectra. Except for the emerging peak at 280 nm and 420 nm, other intrinsic peaks can be found at 395 nm (assigned to yellow chromophore dopamine o−quinone) and 300–475 nm (assigned to aminochrome), respectively [28].

Some researchers also witnessed this kind of interaction. For example, dopamine polymerization with the presence of cinnamaldehyde revealed a new peak at 438 nm in the PDA−cinnamyaldehyde complex [40]. In contrast, dopamine grafting via a EDC/NHS coupling reaction to the gelatin did not generate the peak at 400–420 nm because catechol was not oxidized into quinone [12]. The above results demonstrated the successful polymerization of dopamine on the gelatin chain. XRD analysis showed that both gelatin and Gel−PDA have a wide peak at 21 degrees, which is typical for disordered polymers (Figure 1b). PDA coating did not cause an obvious decrease in the peak strength, implying that PDA coating has little impact on the gelatin crystallinity. At the FT−IR spectra, the peak at 2830 cm^−1^ corresponding to −CH_2_ symmetric stretching of neat gelatin was shifted to 2829 cm^−1^ for Gel−PDA (Figure 1c). The stretching vibration band (3399 cm^−1^) of N−H and O−H groups in the gelatin was diminished to some extent in the Gel−PDA. Meanwhile, the band for C−H stretching vibration exhibited a big shift (from 3140 cm^−1^ in gelatin to 3175 cm^−1^ in Gel−PDA) and was weakened by dopamine polymerization, suggesting that PDA strongly interacted with the functional groups of gelatin. Moreover, a new peak at 1584 cm^−1^, which is corresponds to the N−H bending vibration, also confirmed the PDA presence on the Gel−PDA complex [41]. A similar trend was also witnessed in the other reports. Researchers found that the broad band at 3500–3000 cm^−1^ originally assigned to the stretching vibration of −NH, −OH, and aromatic −CH of dopamine might be weakened rather than belonging to the stretching vibration of free amino groups of PDA after polymerization [42]. In the case of nanocellulose (CNC), it was mentioned that the alteration at 1200–1600 cm^−1^ is mainly associated by the dopamine polymerization. PDA complexation with CNC was identified by some emerging peaks at 1508 cm^−1^ and 1284 cm^−1^, which can be associated with N−H scissoring and C−O stretching, respectively [36].

### 3.2. Fabrication and Characterization of Hydrogel

The formation mechanism was illustrated in Figure 1. The main driving force to form a hydrogel is a borate−didiol interaction between guar gum molecular chains, as well as guar gum and CNC. Interestingly, hydroxyphenol groups of dopamine or tannic acid are also reported to make a complexation with borate, and this fact supported the possible interaction of Gel−PDA and guar gum or CNC [43,44]. Afterward, hydroxyl groups in guar gum or CNC are feasible to form hydrogen bonds with amino or catechol groups in Gel−PDA. Within the Gel−PDA chains, Schiff base or Michael addition might be present between primary amine groups of gelatins and quinones produced from catechols of PDA. Adhesive, self−healing, and tough hydrogel was formed as soon as the borate solution was dropwise added into the polymer mixture dispersions. FT−IR analysis exhibited that all hydrogels have broad peaks around 3412 cm^−1^ and 1635 cm^−1^ corresponding to O−H stretching vibration and C=O stretching vibration, respectively (Figure 2a). The GG+CNC hydrogel had no visible difference from the neat GG hydrogel except for the shifting of the C=O stretching vibration band (from 1631 cm^−1^ to 1611 cm^−1^) because of structural similarities between GG and CNC. This C=O stretching vibration around 1630 cm^−1^ might have originated from the amide bond or quinone groups (oxidation form of catechol in PDA). Therefore, the shifting of this band suggested the CNC complexation with GG during hydrogel synthesis. Meanwhile, an absorption band shown at 1362 cm^−1^ can be assigned to an asymmetric extension of B−OC), confirming the cross−linking between the borate and the catechol group [45]. Gel−PDA addition caused a new peak at 1535 cm^−1^ which is associated with the N−H bending vibration of PDA. The C=O stretching vibration in Gel−PDA+GG+CNC hydrogel shifted from 1631 cm^−1^ to 1662 cm^−1^, implying the probable interaction between Gel−PDA and GG or CNC. The microstructure was observed using SEM. All hydrogels displayed a homogenous and porous network, confirming the formation of typical gel structures (Figure 2b). Neat guar gum had big pores at 50 μm, while that of Gel−PDA+GG+CNC hydrogel was about 30 μm. Moreover, the structure of guar gum hydrogel became significantly tight with the addition of Gel−PDA and CNC, suggesting that cross−linking density might be increased. Previous reports indicated that dopamine polymerization and catechol−based chemistry can improve cross−linking, leading to uniform pore sizes [33]. The higher cross−linking is known to decrease the pore sizes in the hydrogel, while the lower molecular weight of precursor polymer could loosen and enlarge the pore sizes [46].

### 3.3. Rheology, Self−Healing, and Injectability of Hydrogel

The rheological property of hydrogel is very important for biomedical application and the storage modulus (G′), and loss modulus (G″) was measured in two ways: strain sweep and frequency sweep. Within the frequency range of 0.01–100 Hz, the storage modulus (G′) and loss modulus (G″) of neat GG hydrogels was measured. As shown in Figure 3a, G′ of GG was almost the same as G″ at the first step of the frequency range, and this trend was reversed at the remaining steps, finally showing the large difference between the two moduli. Actually, the higher storage modulus than the loss modulus in the whole range is a critical index to determine the viscoelastic property; therefore, neat GG hydrogel prepared in our work might be a weak gel. A similar result was also witnessed by other reporters, who found that borate−guar gum hydrogel prepared at 0.5% consistency has almost the same G′ and G″ values at the low−frequency zone [11]. Considering the relatively high consistency of guar gum (1.5%) in our work, the presence of a low G′ value can be attributed to the difference in the molecular weight and the polymerization of guar gum.

Compared with the GG hydrogel, G′ of GG+CNC, Gel−PDA/GG, and Gel−DA+CNC were clearly higher than G″ across the whole frequency range, implying that these hydrogels possess excellent mechanical properties. The storage modulus of GG hydrogel (152 Pa) was significantly enhanced with the addition of Gel−PDA(267 Pa), while CNC also showed a mechanical reinforcement effect. The highest storage modulus was displayed in Gel−PDA+GG+CNC hydrogel (288 Pa) (Figure 3c). The loss factor, tan δ, is a critical index that determines hydrogel behavior, and it was usually regarded that a lower tan δ reflects a more stable elastomer [13]. In our work, the tan δ of all hydrogel samples was smaller than 1 and weakly frequency dependent, implying a gel−like status. The tan δ of the neat GG hydrogel was 0.29 at the final frequency step, whereas this value was clearly reduced to 0.11 with the CNC (Figure 3d). This result confirmed the critical role of CNC in the stability of GG hydrogel behavior. There was no significant difference in tan δ between GG+CNC, Gel−PDA+GG, and Gel−PDA+GG+CNC hydrogel, implying that they are more stable than GG hydrogel. Especially, Gel−PDA introduction in the GG hydrogel brought the obvious improvement of all rheological indexes, suggesting that Gel−PDA participation in the GG hydrogel is a borate cross−link, not a physical unfolding (because hydrogel preparation was performed at 37 °C). From the strain sweep result, the intersection point of G′ and G″ of GG hydrogel was about 290% of strain (Figure 3b). After this critical strain, G′ of GG hydrogel decreased quickly and started to be lower than G″, meaning that the hydrogel network is damaged and turned into a solution status. With the Gel−PDA and CNC, the intersection did not happen even after 1000% strain, and G′ constantly kept a higher value than G″. This result confirmed the enhancement of mechanical properties of the GG hydrogel by the Gel−PDA and CNC. With regards to the biomedical application of hydrogel, self−healing capacity is also important since many body sites are often subjected to bending, twisting, or other movements, which can easily cause hydrogel damage. Due to the reversible nature of borate cross−linking, GG−based hydrogel showed a fast self−healing capacity [10]. Likewise, GG+CNC, Gel−PDA+GG, and Gel−PDA+GG+CNC hydrogel all exhibited excellent self−healing capacities due to the involvement of Gel−PDA and CNC in the borate−mediated chemistry. For the self−healing test, round−shaped Gel−PDA+GG+CNC hydrogel was cut into two pieces and then put together. After 5 min, the two pieces were reattached and perfectly self−healed without any external stimulus. This reformed piece was stretchable (Figure 4a). Furthermore, the separated two pieces on the finger fully adhered to each other in 10 min when they were brought into contact, and this single piece was not damaged by bending (Figure 4b). Our results confirmed the promising application of hydrogel in the biomedical field for wound healing or biosensors.

When hydrogel is applied at the subcutaneous sites or intestine, injectability can significantly affect its usage. Therefore, good injectability is also an important index of biomedical hydrogels. However, there exists some contradiction between mechanical property and injectability. If the hydrogel does not have a good shear−shining property, it cannot be output from the narrow needle [31]. So, an excessively tough hydrogel is difficult to be injected. In our study, the neat GG hydrogel was hard to inject with the 23−gauge needle despite its low mechanical strength. However, Gel−PDA+GG and Gel−PDA+GG+CNC hydrogel were successively injected through a 23−gauge needle, implying that Gel−PDA can improve the shear−shinning property of the GG hydrogels. Additionally, both hydrogels still kept their fibrous appearance after they were injected under the water (Appendix A).

### 3.4. Tissue Adhesion and Anti−Oxidant Capacity

The tissue adhesive capacity of hydrogels was measured by the lap shear test using wet pig skin, and a macroscopic adhesive test was performed as well. As shown in Figure 5a, the neat GG hydrogel had 18.78 ± 3.58 kPa of lap shear strength, and this adhesive value was increased to 21.71, 29.74, and 35.23 kPa for GG+CNC, Gel−PDA+GG, and Gel−PDA+GG+CNC hydrogel, respectively. CNC probably does not cause the visible improvement of adhesion to pig skin, whereas Gel−PDA brought a great difference in tissue adhesion. It was reported that the polymerization reaction in the alkaline environment is similar to the catechol oxidation progress in mussel−secreted proteins under seawater, and this process endows the mussel with very high adhesion because it can protect the “adhesive” catechol groups by partially oxidizing into ο−quinones in this way [47]. Abundant catechol groups of PDA are easily deprotonated to become reactive quinone groups, and then consequentially interact with amine, thiol, or imidazole groups of pig skin proteins via the Michael−type addition and Schiff−base reaction [32].

Additionally, some physical interactions originating from catechol groups including hydrogen bonds, π–π stacking, and π–cation interactions can contribute to the good adhesion of the hydrogel. All hydrogel samples showed significantly higher values of tissue adhesion compared with commercial dressing (around 5 kPa), confirming their potential usage as tissue adhesives (Appendix A). Excessive production of reactive oxygen species (ROS) in the damaged tissue is one of the critical obstacles to hinder quick recovery. In this sense, the anti−oxidant activity of biomedical hydrogels to remove ROS can determine its treatment efficiency. As shown in Figure 5c, the DPPH free radical scavenging ratio of GG+CNC hydrogel did not have a significant difference from that of GG hydrogel after 1 h of incubation. The anti−oxidant capacity of Gel−PDA+GG and Gel−PDA+GG+CNC hydrogel was increased by about five times, suggesting that catechol groups in the PDA can obviously improve the anti−oxidant activity. The improvement of the ROS scavenging ratio by the catechol group was also mentioned in previous papers [48,49].

### 3.5. Hemostasis, Hemolysis, and Cytotoxicity of Hydrogels

Hemostasis is the crucial property of biomedical hydrogel when it is used as wound dressings. With the rapid stop of bleeding, hydrogel dressing can exert good wound closure. The hemostatic capacity of hydrogel was measured using a rat hepatic hemorrhage model. As shown in Figure 6a, compared with the bleeding of 149.33 mg in the control group, GG and GG+CNC hydrogel showed obviously small bleeding (28.4 and 25.16 mg, respectively). Moreover, the blood loss of GG hydrogel was further decreased with the Gel−PDA addition to a great extent (19.56 and 14.16 mg for Gel−PDA+GG and Gel−PDA+GG+CNC, respectively). This result is well−matched with the rheology and adhesion of hydrogels. Previously, it was already reported that good adhesion and mechanical strength enable the hydrogel to adhere to the injury sites quickly to seal the bleeding. Meanwhile, amino groups of Gel−PDA were found to stimulate erythrocyte aggregation. In addition, the vasoconstrictive activity of dopamine is helpful to reduce blood loss because the blood vessels can be constricted rapidly [50]. A hemolysis assay was performed to clarify the possible effect of hydrogels on the red blood cells (RBCs), and there was no significant difference between hydrogel samples (Figure 7a). Considering that the hemolysis ratio limit for biomaterials is below <5%, all hydrogels exhibited good hemocompatibility, suggesting their medical safety.

The cytocompatibility analysis was performed using a CCK−8 assay kit for hydrogel extract liquid, and the result showed that all hydrogel samples have no clear cytotoxicity (Figure 7b). Guar gum and nanocellulose are all nontoxic biopolymers, while dopamine inspired by mussels is also a safe substance with low cytotoxicity, so it is extensively used for the bioactive modification of tissues. Other researchers also demonstrated similar results. GG hydrogel even showed higher cell viability than the control (DMEM medium), suggesting the cell proliferative effect. Additionally, the gel−dopamine introduction in the GG hydrogel also caused the improvement of cell growth after 24 h. However, the Ag loading into the gel−dopamine reduced the cell viability to some extent [12]. In another paper, silk fibrion PDA hydrogel had a higher cell density than culture medium or silk fibrion hydrogel. Nonetheless, the higher consistency of dopamine was not preferable for cellular growth because it might inhibit cell activity [42]. So, a slight decrease in the cell viability of Gel−PDA+GG+CNC hydrogel is probably attributed to the higher dopamine concentration.

## 4. Conclusions

In summary, we prepared functional hydrogels which can be used as wound dressing and tissue scaffolding. The Gel−PDA and CNC complexation successively overpassed the shortcomings of guar gum hydrogel. PDA coating on the gelatin is a facile and effective method to improve the gelatin property, leading to the proper involvement of gelatin in the borate−based guar gum hydrogel. Meanwhile, abundant hydroxyl groups of CNC enabled its contribution to the stable hydrogel formation. Therefore, tissue adhesion and rheology of GG hydrogels are obviously enhanced. Additionally, Gel−PDA+GG+CNC hydrogels had good mechanical strength, high ROS scavenging activity, acceptable hemocompatibility, and cytotoxicity. Our study confirmed that PDA coating chemistry can be a promising approach to modifying the hydrogel precursor polymers, broadening its application in the biomedical field.

## Data Availability

All related data and methods are presented in this paper and the supplementary materials. Additional inquiries should be addressed to the corresponding author.

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
