# Peer review of "Functional Enhancement of Guar Gum−Based Hydrogel by Polydopamine and Nanocellulose"

_foods, 2023, doi:10.3390/foods12061304_

Round 1

Reviewer 1 Report

The material seems very interesting. Nevertheless several remarks/comments:

The paper is constructuted in a way so the experiment may be reproduced severeal data are missing though:

 a) line 98 - freeze drying conditions

 b) line 101 - what kind of spectroscopy is being used (reflection, transmission, absorbtion) - it can be deduced from figures nonetheless plain text would be helpfull, have you used any additional modules for the spectrometer?

 c) SEM - what was sample preparation procedure (gold, platinum covering?), what was the maginfication?

 d) line 206 - on what grounds do you indentify 280-290 nm peak as for gelatine? is there any other experimental result confirming this claim?

e) in order to fully reproduce the results the ANOVA tables together with experimental results should be introduced. At least ANOVA tables. Why there is no ANOVA analysis in Figure 6 if previous results has been analysed?

Figure 6b seems be totally different format that others, should be standarized for clarity and consistency.

Author Response

Dear reviewer!

Reviewer 2 Report

I find the study really innovative and interesting. There are some questions and comments, which can be helpful in enhancement of the manuscript. Besides, I suggest to revise the manuscript in terms of language and style of text.

·       Ln 18: Grammar check.

·       The significance of the study was to develop a green hydrogel, using natural polymers. Although the base of the final hydrogel is from natural polymers, it has been chemically cross-linked. You could call it a bio-, green hydrogel, when the cross-linking agent has also been from bio-resources (like fungal enzymes).

·       Improve the Keywords.

·       Ln 26: As you are using hydrogels with different sources, it would be better to define various types of hydrogels in the introduction section.

·       Ln 38: A molecular structure of Guar gum and the cross-linked Guar hydrogel will more advantageously clarify the paragraph. In addition, it would be more informative to explain the reason behind the weak structure of cross-linked guar hydrogel.

·       Ln 72: You should explain the aim of the work in last paragraph of the introduction, NOT the whole method, results and significances of the study. This paragraph should be totally revised.

·       Ln 91 & 92: Grammar check.

·       Ln 131: Have you also measured Linear Viscoelastic Region?

·       Ln 152: Why have you repeated the test five times? Three times was not enough?!

·       The references of all the Methods are missed. It is necessary to add the reference of each method and assay in the “Material and Methods” Section.

·       Ln 231: Results should be more discussed. Please add a discussion part.

·       Ln 250: where is the rest of the sentence?

·       Ln 255: Results should not be explained in the title of figure. Title of Scheme 1 should just include the figure description in the short sentence.

·       Figure 3 “a” and “b”: Graphs are not clear, signs are too small and colours are not detectable.

·       Ln 297: Have you also measure loss factor? It is interesting to know, how close tanδ to 0 in the final gels was.

·       How much was Storage Modulus of different gels? Did GG+Gel-PDA+CNC have the highest G’? It would be informative to have a table including G’, G’’ and tanδ of the gels to see how stronger the cross-linked hydrogels are.

·       Section 3.3: Discussion of the results should be improved.

·       Lines of the whole text are not perfectly justified. Please adjust all the lines in order to distribute the text between the margins.

·       Ln 345: Grammar check.

·       Ln 374: Grammar check.

·       Ln 374: Discussion of the results should be improved. You should support your own results in the “Conclusion” section.

·       Figure 6 should be inserted before the Conclusion.

·       Enhance the references of the study by adding the references of “Material and Methods” and improving “Discussion of the results”.

Author Response

Dear reviewer!

Reviewer 3 Report

Paper foods-2229724 refers to the use of food components in the form of hydrogels for biomedical applications. 1. As presented, the manuscript does not accurately reflect the essence of the Special Issue "Application of Food Hydrocolloids for Hydrogels and Packaging" as well as the main scope of Foods. The authors should eliminate this significant shortcoming in the introduction and discussion. 2. Several polymer components are used in the work, but their characteristics, in particular, molecular weights, are not available. It probably matters for the processes being developed. 3. Why is nanocellulose used in research rather than the more readily available and cheaper microcrystalline cellulose or the recently popular bacterial cellulose? 4. Need to improve the quality of Scheme 1 and Figure 2. 5. Carefully check the design of the manuscript. For example breaks in the text (lines 250-257 and 305-309, as well as in other places).

Author Response

Dear reviewer!

Reviewer 4 Report

The manuscript entitled Functional enhancement of guar gum-based hydrogel by poly-dopamine and nanocellulose was evaluated. The manuscript has many corrections to be made, the methodologies to be better explained and some inconsistencies in the sampling and in the results. Please, find some of the suggested corrections pointed bellow:

1. I reccomend the authors to insert a fluxogram or a scheme that represents the step-by-step of the synthesis of Gel-PDA and the Gel-PDA-Guar gum-CNC hydrogel. This will help to better understanding of the proposed work.

2. In item 2.2 Synthesis of Gel-PDA, you do not cite how many samples have you prepared. Did you prepared only one sample? Or did you prepared a triplicate of this material? Please insert the information regarding sampling.

3. In item 2.3 Characterization of Gel-PDA, the used techniques should be better described. In this context, the methods could be presented as in subtitens, such as 2.3.1 Ultraviolet-visible (UV-VIS) spectrometry; 2.3.2. Fourier Transform Infrared (FTIR) spectroscopy and 2.3.3 X-ray diffraction (XRD) analysis. In the UV-Vis analysis, explain why did you choose the range of 200-700nm, and why not a wider range starting at 100 nm to 800nm.

4. In item 2.4 Fabrication of Gel-PDA+Guar gum+CNC hydrogel, once again, you do not cite how many samples have you prepared. Did you prepared only one sample? Or did you prepared a triplicate of this material? Please insert the information regarding sampling.

5. I have noted that none methods used to prepare and characterize the samples cite any liteture. Please, revise the text and insert the references of each methodology used.

6. The authors do not informed the aproval in the appropriate Ethics Committee autorization/registration for the use of animals in this work, such as in item 2.5.5. Hemolysis and hemotasis assay. The instruction for authors of MDPI cites that the "Authors must include details on housing, husbandry and pain management in their manuscript. For further guidance authors should refer to the Code of Practice for the Housing and Care of Animals Used in Scientific Procedures [2], American Association for Laboratory Animal Science [3] or European Animal Research Association [4]. (https://www.mdpi.com/journal/foods/instructions)". Please, verify the ethics requirements for your manuscript and add the proper information regarding it.

7. The quantity of animals (Sprague Dawley rats) used to these assays is missing. Information regarding the time living (days or months?) of these animals is also missing. After performing the hemolysis and hemostasis assays in the Sprague Dawley rats, did any of these animals died? Did you follow any literutures or international protocols to perform these experiments? If yes, please, insert the information. Despite the application of anesthesy, did you used any other method to reduce/minimize the harm to animals? As the ethics guidelines of MDPI requires that the "Authors should particularly ensure that their research complies with the commonly-accepted '3Rs: Replacement of animals by alternatives wherever possible; Reduction in number of animals used, and Refinement of experimental conditions and procedures to minimize the harm to animals".

8. In the item 2.5.6. Cytotoxity assay, you used frozen HepG2 cells. Please insert the information regarding the source (scientific research or commercial institutions) of these cell lines, as well as the Ethical Statements regarding it, according to the https://www.mdpi.com/journal/foods/instructions. 

9. In the item 2.5.6. Cytotoxity assay, you cited that 'Each sample had a six-wells treatment in parallel'. In this context, how many samples did you tested? You used a hydrogel extract, however you do not describe how did you obtained this extract? It was an aqueous or DMEM medium based extract? Please, insert a reference literature for the method used in this assay.

10. In the item 3.1. PDA coating on the gelatin matrix, you cite that 'Dopamine with active catechol group is easily oxidized at the alkali condition and further polymerized to form a polydopamine'. How can you infer this whitout show any result of a physico-chemical analysis of the oxidation reaction to form the polymerized form of Dopamine? Please, insert literature or results that support this statement.

11. In the item 3.1., you cite that 'UV-VIS spectrum showed an intrinsic peak of gelatin at 280-290 nm'. However, I do not observed a peak at 280-290nm. I noted that possibly your UV-Vis spectra had a peak before 250nm, as the absorbance decrease from 250 to 280nm, where it exhibits a plateau from 260-280nm. I believe you could perform this analysis again and verify if you have any peak below 250nm.

12. In the item 3.1. (lines 209-210), you cite that 'a weak peak shown around 420 nm is an an other evidence for the PDA complexation with gelatin, which is time-dependent'. However your results does not show this weak peak at 420nm. Then, later in the paragraph you explain that this peak was not present in your results because catechol was not oxidized into quinone. These statements are confusing the understanding of your work. Afterwords, you state that these results demonstrated the polymerization of dopamine into the gelatin chain, which is also confusing. You should explain with supporting literature how the polymerization reaction of dopamine on the gelatin chains occurs.

Author Response

Dear reviewer!

Round 2

Reviewer 3 Report

Thanks to the authors, you have done a great job. The review has been significantly improved.

Reviewer 4 Report

No further comments.